# Research on a Particle Filtering Multi-Target Tracking Algorithm for Distributed Systems

**DOI:** 10.3390/s25113495

**Published:** 2025-05-31

**Authors:** Bing Han, Zilong Ge, Zhigang Su, Jingtang Hao

**Affiliations:** Sino-European Institute of Aviation Engineering, Civil Aviation University of China, Tianjin 300300, China; b-han@cauc.edu.cn (B.H.); 2023122005@cauc.edu.cn (Z.G.); jthao@cauc.edu.cn (J.H.)

**Keywords:** multi-objective tracking, coupled measurements, data fusion preprocessing, optimal particle weights

## Abstract

The growth of unmanned aerial vehicle applications in the low-altitude economy demand advanced multi-target tracking systems. Unlike traditional approaches that assume independent measurements, distributed systems generate coupled measurements containing additional target relationship information. This paper proposes a novel distributed particle filtering algorithm through introducing the coupled measurement into the conventional particle filtering method. In the proposed method, we fuse direct and coupled measurements via optimization and then build a cost function to optimize the particle weights. Comparative evaluations across motion models, noise levels, and the number of targets demonstrate the outperforming performance of the proposed method compared to conventional particle filtering and the unscented Kalman filtering algorithm, with more than 7% accuracy improvement over baselines. The results prove particular robustness to measurement noise and the increasing number of targets.

## 1. Introduction

The rapid development of the low-altitude economy has led to the widespread deployment of unmanned aerial vehicles (UAVs) across diverse domains including logistics, precision agriculture, infrastructure inspection, and emergency response operations. However, the exponential growth in UAV fleets and increasingly complex operational environments introduce significant safety challenges, particularly regarding signal interference, loss-of-control incidents, and thus leading to collision risks. Therefore, operational safety has become a key challenge for UAV applications. Real-time and accurate multi-target tracking technology can simultaneously monitor the status, position, and trajectory of multiple UAVs and thus can effectively avoid air traffic conflicts and improve overall operational efficiency and safety. Specific multi-target tracking research can not only optimize the performance of the UAV management system but also provide a technical guarantee for the sustainable development of low-altitude economy and promote the expansion of UAV applications to wider and safer fields.

For the tracking problem, classical tracking methods, including the Kalman filter (KF) [1,2], extended Kalman filter (EKF) [3,4], unscented Kalman filter (UKF) [5,6], and particle filter (PF) [7,8], are commonly used for target tracking. PF methods use a probability model in place of a determined motion model and thereby show good tracking capability for nonlinear moving targets. In order to accurately distinguish and continuously track multiple targets, many scholars have worked on reducing noise interference to minimize the complexity of radar multi-target data association. In 2003, R.P.S. Mahler [9] approximated the recursive first-order statistical moments of multi-target probability density and shifted the multi-target tracking problem to a single-target state space to solve it, which can prevent the complex data association problem in multi-target tracking. In 2013, in order to reduce the multi-path propagation effect in radar on multi-target tracking, B. Habtemariam [10] proposed a multi-detection joint probabilistic data association filter algorithm for multi-target tracking, and it was capable of forming the association probability of multiple target detections to evaluate the likelihood of measuring the true value of target measurements and improving the multi-target state estimation accuracy. In 2015, the gated strategy was introduced for target state space measurement likelihood [11] for the overall confidence level of the probability hypothesis density filter to realize the improvement of the target tracking accuracy through the reduction in clutter interference. In 2020, for the dense clutter and the interference of multi-range false targets on multi-target tracking, the target amplitude information and phase quantization information were used to distinguish the targets and restrain dense clutter [12], which obtained a better correct discrimination rate of the multi-range false targets and high-accuracy tracking performance.

In radar surveillance systems, the error caused by measurement system noise is also an important factor that degrades the multi-target tracking performance of the surveillance system. To improve data processing technology and tracking accuracy, in 2018, the classical linear KF was applied to track the pseudo-linear state-space model [13] and compensate for measurement bias, and reduced system bias and improved multi-target tracking accuracy. In the same year, Cai-ling Wang [14] proposed iteratively updating the measurement process and combining it with Gauss–Newton iteration theory, which further improved the Kalman filtering accuracy of the automotive radar target tracking algorithm. In 2019, the Pareto optimization theory was introduced to deal with dwell time of the phased array radar [15], a linear programming method combined with the Min–Max algorithm, which improved simultaneously the multi-target tracking accuracy, as well as the target signal-to-noise ratio. In 2021, for the radar measure information redundancy problem, a sparsity-enhanced sequential convex programming approach [16] was proposed to perform measure information selection and power joint optimization allocation, which improved multi-target tracking accuracy in the presence of sensor position uncertainty. In 2022, Su [17] proposed a joint optimization algorithm for grouped, centralized, Multiple-Input–Multiple-Output (MIMO) radar waveform control and resources in a maneuvering target tracking scenario. The algorithm took minimizing the maneuvering target tracking error as the optimization objective and used an improved particle swarm algorithm to solve the problem, which achieved the purpose of improving the maneuvering target tracking accuracy.

The above research results have laid a solid foundation for improving the accuracy of multi-target tracking. However, with the expansion of the monitoring range of the surveillance system, the application of distributed surveillance system has been increasing. Multiple mutually independent sensor nodes in the distributed system perform data acquisition, processing, and exchange and fusion through the network. Each node is capable of monitoring and tracking targets within a specific region, while the complementary nature of multi-source data further improves multi-target tracking precision. Owing to these advantages, the superior performance of distributed systems has received increasing attention in the field.

In distributed systems, relative positions [18] and the received signal strength [19] between multiple targets are used for distributed localization in multi-vehicle formations and are effective in reducing the matching complexity in multi-target tracking. In 2021, Wenling Li et al. [20] introduced the coupled information obtained by the sensors into the distributed Kalman filtering system. The boundedness of the estimation error is discussed and guaranteed. In [18,20], the coupled measurement information is discussed to reduce tracking errors. However, the target tracking process is based on the Kalman method, which is not efficient for nonlinear multi-target motion scenarios. In [21], the authors proposed a near-optimal, distributed implementation of the particle filter for large-scale dynamical systems with sparse measurements, which can provide significant computational savings. In order to promote efficiency in the use of localized information, a distributed nonlinear estimation method based on particle filtering was proposed later [22]. This method achieved efficient nonlinear state estimation in distributed networks by combining Gaussian approximation and consensus propagation algorithms. In order to improve the information exchanging efficiency, the authors of [23] proposed the approximating local likelihood function by Gaussian distribution and by exchanging only the mean and covariance. The method significantly reduced the inter-sensor communication overhead and power consumption while maintaining tracking accuracy. In [19,21,22,23], the target tracking process is suitable for the nonlinear targets, but the coupled measurements between different sensors are not discussed.

All of these studies have demonstrated the superior tracking performance of multi-target tracking based on distributed systems. However, the distributed particle filtering problem has not been investigated for multitarget tracking systems with coupled measurements. In this paper, the coupled measurements of the sensors in the distributed system are introduced in the conventional particle filtering algorithm. By constructing the cost function based on coupled measurement and direct measurement, we solve the particle weights in particle filtering and complete the multi-target state estimation and tracking algorithm. The proposed distributed particle filtering algorithm can effectively filter out the measurement noise and improve the multi-target tracking accuracy.

The main technical contributions of this paper can be summarized as follows:The integration of coupled measurements into distributed particle filtering to enhance multi-target tracking accuracy.The formulation of a cost function to optimize particle weight distribution through constrained likelihood maximization.A fused measurement to minimize the measurement noise.

The structure of this paper is as follows. In Section 2, we provide an introduction to the model of the multi-target tracking problem. In Section 3, we describe the theoretical computation method of particle weight calculation, and in Section 4, we present the results of numerical simulations that assess the performance of our proposed algorithm. We compare its performance with several classical methods commonly used in target tracking, such as conventional particle filtering and unscented Kalman filtering. In Section 5, we conclude the paper with a summary of our findings and potential future research directions.

## 2. Problem Statement

To describe the motion and tracking process of a target, we assume that the motion and measurement for any nonlinearly moving target *i* can be expressed as(1)$$\begin{array}{c} \begin{array}{cc} x_{i,k} & =f\left(x_{i,k-1}\right)+w_{i,k}\\ z_{i,k} & =h\left(x_{i,k}\right)+v_{i,k} \end{array} \end{array}$$
where $x_{i,k}$ denotes the state of target *i* at time *k*, $z_{i,k}$ denotes the measurement of target *i* at time *k*, f(·) denotes the state transfer function of the moving target, and h(·) denotes the measurement function. $w_{i,k}$ stands for the process noise, and $v_{i,k}$ is the measurement noise of the system; both are Gaussian noises with a 0 mean, and the covariance matrices are Q and R, respectively.

For the distributed system, in order to eliminate the systematic error between different measurements, we subtract the measurements in pairs for the *m* targets and introduce the coupled measurements between different targets based on the same sensor, and the coupled measurements can be expressed as follows:(2)$$\begin{array}{c} \begin{array}{cc} z_{i,j,k} & =h\left(x_{i,k}\right)-h\left(x_{j,k}\right)+v_{i,j,k}\\ & =H\left(x_{i,k},x_{j,k}\right)+v_{i,j,k}\;\;i\neq j\in \left[1,2,\ldots ,m\right] \end{array} \end{array}$$
where H(·) denotes the measurement function for the matched target pair, and $v_{i,j,k}$ denotes the systematic measurement noise for this sensor, which is also a 0-mean Gaussian noise with a covariance matrix R, since it is the same measurement system. Normally, the covariance matrix R is a little different for each sensor for multiple reasons, such as the distance between sensor and target, the signal transmission link, etc. Since we mainly discuss the effect of the coupled measurement in this paper, we suppose that all the sensors have the same covariance matrix R. For the reader’s convenience, an example with four targets is depicted in Figure 1. Since there are many coupled measurements that can be matched under the same sensor, resulting in information redundancy, we select only one pair of targets for each sensor to calculate the coupled measurements. After we number all the targets, the coupled measurements can be presented as(3)$$\begin{array}{c} z_{i,i+1,k}=H\left(x_{i,k},x_{i+1,k}\right)+v_{i,i+1,k} \end{array}$$
where $x_{m+1,k}$ is set to $x_{1,k}$.

Thus, the state estimation of the tracked target is expressed as computing the expectation $E({\hat{x}}_{i,k}|z_{i,1:k},z_{i,i+1,1:k})$ of the state of the tracked target *i*. In order to compute the expectation, we need to first obtain the probability density function $p(x_{i,k}|z_{i,1:k},z_{i,i+1,1:k})$. Since the probability density function is difficult to estimate directly, particle filters have been introduced, whose basic idea is to utilize a large number of particles ${\left\{x_{i,k}^{n}\right\}}_{n=1}^{N}$ to approximate the matched density function:(4)$$\begin{array}{c} \hat{p}\left(x_{i,k}|z_{i,1:k},z_{i,i+1,1:k}\right)=\sum_{n=1}^{N}\omega_{i,k}^{n}\delta_{x_{k}^{n}}\left(x_{k}\right) \end{array}$$
where $\omega_{i,k}^{n}$ is the weight corresponding to each particle $x_{i,k}^{n}$ and $\delta_{x}\left(\cdot \right)$ denotes a particle sampling about state $x_{i,k}$.

In conventional particle filtering, the particle weights $\omega_{i,k}^{n}$ are determined based only on the predicted values of the target state and the measurements of the target state as follows:(5)$$\begin{array}{c} \omega_{i,k}^{n}\propto \rho \left[z_{i,k}-h\left(x_{i,k|k-1}^{n}\right)\right] \end{array}$$
where ρ(·) denotes the probability density function of the measurement noise. After the introduction of the coupled measurements, the Formula (Equation 5) for calculating the particle weights is no longer applicable. Therefore, we need to solve the particle weight distribution for target tracking according to the direct measurements $z_{i,k}$ and coupled measurements $z_{i,i+1,1:k}$ and then complete the target tracking based on the particle filtering algorithm.

## 3. Distributed Particle Filtering

Our goal is to establish the target tracking particle weights based on the direct and coupled measurements, as well as the predicted values of the target states based on the distributed particle filtering system. Since both the direct and coupled measurements contain certain noise, in order to make the weight estimation more accurate and the state estimation error lower, before calculating the particle weights, we fused the acquired direct and coupled measurements to obtain the fused measurements. Subsequently, we established a particle weight solving method based on distributed particle filtering system.

### 3.1. Measurements Fusion

For *m* targets to be tracked, we define(6)$$\begin{array}{c} \begin{array}{cc} z^{r} & ={\left[z_{1}^{r},z_{2}^{r},\ldots ,z_{m}^{r}\right]}^{T}\\ z & ={\left[z_{1},z_{2},\ldots ,z_{m}\right]}^{T}\\ z_{d} & ={\left[z_{1,2},z_{2,3},\ldots ,z_{m,1}\right]}^{T}\\ \hat{z} & ={\left[{\hat{z}}_{1},{\hat{z}}_{2},\ldots ,{\hat{z}}_{m}\right]}^{T} \end{array} \end{array}$$
where ${\left(\cdot \right)}^{T}$ denotes the transpose operation, $z^{r}$ denotes the noise-free ideal measurement of *m* targets under the systematic measurement function h(·), *z* denotes the actual direct measurement, $z_{d}$ denotes the coupled measurement, and $\hat{z}$ stands for the fused measurement. From Equation (Equation 2), the coupled measurement can be expressed by two ideal measurements and noise calculation; then, we can get(7)$$\begin{array}{c} \begin{array}{cc} z_{d} & =A_{m}z^{r}+{\left[v_{1},\ldots v_{m}\right]}^{T}\\ A_{m} & =\left[\begin{array}{ccccc} I_{s} & -I_{s} & \ldots & 0 & 0\\ 0 & I_{s} & \ldots & 0 & 0\\ \ldots & \ldots & \ldots & \ldots & \ldots \\ 0 & 0 & \ldots & I_{s} & -I_{s}\\ -I_{s} & 0 & \ldots & 0 & I_{s} \end{array}\right] \end{array} \end{array}$$
where $I_{s}$ denotes the unit matrix and *s* denotes the dimension of the measurement of one sensor. From Equations (Equation 1) and (Equation 2), we can obtain(8)$$\begin{array}{c} \begin{array}{cc} E(z_{d}-A_{m}z^{r}) & =E(z-z^{r})=0\\ E\;\;\;[\;\;\;(z_{d}-A_{m}z^{r})(z_{d}-A_{m}z^{r}{)}^{T}\;\;\;]\;\; & =E\;\;\;\left[\;\;\;(z-z^{r})(z-z^{r}{)}^{T}\;\;\;\right]\;\;=\mathrm{blkdiag}(\underset{m}{\underset{︸}{R,..,R}}) \end{array} \end{array}$$
where blkdiag(·) is the block diagonal function.

To estimate the optimal fused measurement, we construct a cost function representing the errors between true measurement and direct measurement, as well as between true measurement and coupled measurement:(9)$$\begin{array}{c} \begin{array}{c} J\left(\hat{z}\right)={\left(z-\hat{z}\right)}^{T}\left(z-\hat{z}\right)+{\left(z_{d}-A_{m}\hat{z}\right)}^{T}\left(z_{d}-A_{m}\hat{z}\right) \end{array} \end{array}$$

The optimal fused measurement will minimize the cost function, which is obtained by deriving $\hat{z}$(10)$$\begin{array}{c} \begin{array}{c} \frac{dJ(\hat{z})}{d\hat{z}}=2\hat{z}-2z+2A_{m}^{T}A_{m}\hat{z}-2A_{m}^{T}z_{d} \end{array} \end{array}$$

It can be proved that the matrix $I+A_{m}^{T}A_{m}$ is invertible, so the optimal fused measurement can be obtained:(11)$$\begin{array}{c} \begin{array}{c} \hat{z}={\left(I+A_{m}^{T}A_{m}\right)}^{-1}\left(A_{m}^{T}z_{d}+z\right) \end{array} \end{array}$$
and it is easy to prove that the fused measurement is also unbiased. For the covariance, we can get(12)$$\begin{array}{c} \begin{array}{c} E\;\;\;\left[\;\;\;(\hat{z}-z^{r})(\hat{z}-z^{r}{)}^{T}\;\;\;\right]\;\;\;=\mathrm{blkdiag}\;\;\;\left\{\;\;\;\underset{m}{\underset{︸}{R,..,R}}\;\;\;\right\}\;\;\;{\left(I+A_{m}^{T}A_{m}\right)}^{-1} \end{array} \end{array}$$
where the matrix ${\left(I+A_{m}^{T}A_{m}\right)}^{-1}$ is a symmetric matrix and it has all the same elements on the diagonal. It can be proved that the diagonal element is between 0 and 1, and it is only related to the number of targets *m*. The value of the diagonal element decreases with increasing number of targets, so it converges gradually.

### 3.2. Update of Particle Weights

For particle sampling, the unscented sampling method [24] is implemented. The number of sampling points is determined by the dimension of the state, which requires only a small number of sampling points to approximate the state distribution with high computational efficiency. In distributed systems, the information of coupled measurements cannot be extracted directly based on Equation (Equation 5). Therefore, we establish the particle weight relationship of coupled measurements according to the coupled measurements:(13)$$\begin{array}{c} \begin{array}{c} \omega_{i,i+1,k}^{pq}\propto \rho \left[z_{i,i+1,k}^{pq}-H\left(x_{i,k|k-1}^{p},x_{i+1,k|k-1}^{q}\right)\right] \end{array} \end{array}$$
where $\omega_{i,i+1,k}^{pq}$ denotes the particle weights of particle $x_{i,k}^{p}$ and particle $x_{i+1,k}^{q}$ for the coupled measurements, and $z_{i,i+1,k}^{pq}$, ρ(·) denotes the probability density function of the measurement noise. We assume that the targets are all moving independently of each other, so the two particles in each pair are also independent of each other. In noiseless conditions, this assumption leads to the result that the coupled measurement is equal to the difference between two direct measurements, and the expectation of coupled particles equals the difference between two expectations of particles set for two corresponding targets. As the coupled measurements depend independently on two different particles, the following can be obtained:(14)$$\begin{array}{c} \begin{array}{c} \omega_{i,i+1,k}^{pq}=\omega_{i,k}^{p}\omega_{i+1,k}^{q} \end{array} \end{array}$$
where $\omega_{i,k}^{p}$ denotes the weight of particle $x_{i,k}^{p}$, and $\omega_{i+1,k}^{q}$ denotes the weight of particle $x_{i+1,k}^{q}$. However, solving particle weights $\left\{\omega_{i,k}^{n}\right\}$ by using Equation (Equation 13) alone cannot give a unique solution, since the number of paired particles is much larger than the number of particles. Therefore, we propose a cost function for the coupled measurements and the direct measurements:(15)$$\begin{array}{c} \begin{array}{c} Q\left(\omega \right)=\alpha ||\sum_{i=1}^{m}(\omega_{i}Z_{i,i+1}\omega_{i+1}^{T}-z_{i,i+1}){\left|\right|}^{2}+\left(1-\alpha \right)||\sum_{i=1}^{m}(\omega_{i}Z_{i}-{\hat{z}}_{i}){\left|\right|}^{2} \end{array} \end{array}$$
where ω denotes the weight vector consisting of all particles of all targets, $\omega_{i}$ is the particle weight vector for the target *i*, $Z_{i,i+1}=\left[z_{i,i+1}^{pq}\right]$ is the matrix of particle coupled measurements for the target *i* and target i+1, $Z_{i}={\left[z_{i}\right]}^{T}$ is the matrix of measurements for target *i*, and α is the scaling factor. The first part of this cost function represents the difference in coupled measurements between particle prediction and the measurements. The second part represents the difference between the particle and the fused measurements. Both parts represent the error caused by the noise of the measurement system, and our goal is to find the particle weight values that minimize the cost function:(16)$$\begin{array}{c} \begin{array}{c} \omega =\underset{\omega }{\arg}\;\min Q\left(\omega \right) \end{array} \end{array}$$

Since the particle weights should also follow the magnitude relationship shown in Equation (Equation 5), Equation (Equation 5) is transformed into constraints for the optimization problem as(17)$$\begin{array}{c} \begin{array}{c} \rho \left[z_{i,k}-h\left(x_{i,k|k-1}^{p}\right)\right]<\rho \left[z_{i,k}-h\left(x_{i,k|k-1}^{q}\right)\right]\Rightarrow \omega_{i,k}^{p}<\omega_{i,k}^{q} \end{array} \end{array}$$

From Equation (Equation 15), we can see that the objective function is composed of two Euclidean distances, so it is a convex objective function. The restrictive condition Equation (Equation 17) only defines the upper limit for each variable. For possible weight solutions $\omega_{1}$, $\omega_{2}\in \left[0,1\right]$, we also have $\lambda \omega_{1}+\left(1-\lambda \right)\omega_{2}\in \left[0,1\right]$, which proves the convex restrictive condition. As a result, we can use the interior point method to solve this optimization problem. For the computational complexity, the interior point method is a polynomial algorithm that does not introduce much computational load. The total computational complexity of the proposed method is $O(mNd-{\left(mN\right)}^{7/2}\log\left(\epsilon \right))$, where the *m* is the target number, *N* is the particle number, *d* is the dimension of state vector, and the ϵ is the optimization solving accuracy. Compared to the computational complexity of the PF method, O(mNd), and the UKF method, $O(md^{3})$, the proposed method increases the computational load, but it is still a polynomial.

After completing the solution of particle weights, target state estimation can be carried out to complete the particle filtering computation process. Therefore, the complete flow of the distributed multi-target tracking algorithm based on particle filtering explored in this paper is shown in Figure 2. Following initialization, the tracking process proceeds iteratively: first, the target state is predicted via particle sampling. Subsequently, measurements are acquired and target association is performed based on the direct measurements. And data association in multi-target tracking is performed based on the criterion of minimal Euclidean distance between the predicted state estimate and the direct measurement. When using this simple association strategy, a better state estimation can directly result in a better tracking performance with limited measurement noise, and the effect of the various trajectories on the tracking performance can be obvious so that we can directly evaluate the effect of introducing of coupled measurement. The measurements are then fused, and particle weights are updated based on the likelihood between predictions and measurements. Finally, the target state is updated, and the algorithm proceeds to the next iteration. Since we focus on solving the particle weights, we do not consider the disappearance and reappearance of the target.

## 4. Simulation and Analysis of Results 

### 4.1. Simulation Setting

To comprehensively assess the proposed algorithm’s performance and validate the technical contributions, we compares the performance of the proposed method with the conventional particle filtering (PF) algorithm, the unscented Kalman filtering (UKF) algorithm under different simulation conditions. These two methods are both well-known algorithms with excellent performance. Three motion models are considered to represent different motion scenarios in the simulation experiments: the constant velocity (CV) model, the constant turn rate (CT) model and the damped parabolic (DP) model. The target moving models for the three models are presented, respectively, as(18)$$\begin{array}{c} \begin{array}{c} x_{i,k}=\left[\begin{array}{cccc} 1 & T & 0 & 0\\ 0 & 1 & 0 & 0\\ 0 & 0 & 1 & T\\ 0 & 0 & 0 & 1 \end{array}\right]x_{i,k-1}+w_{i,k} \end{array} \end{array}$$(19)$$\begin{array}{c} \begin{array}{c} x_{i,k}=\left[\begin{array}{cccc} 1 & \frac{\sin(\Omega T)}{\Omega } & 0 & \frac{\cos(\Omega T)-1}{\Omega }\\ 0 & \cos(\Omega T) & 0 & -\sin(\Omega T)\\ 0 & \frac{1-\cos(\Omega T)}{\Omega } & 1 & \frac{\sin(\Omega T)}{\Omega }\\ 0 & \sin(\Omega T) & 0 & \cos(\Omega T) \end{array}\right]x_{i,k-1}+w_{i,k} \end{array} \end{array}$$(20)$$\begin{array}{c} \begin{array}{c} x_{i,k}=\left[\begin{array}{cccc} 1 & \frac{e^{\beta T}-1}{\beta } & 0 & 0\\ 0 & e^{\beta T} & 0 & 0\\ 0 & 0 & 1 & \frac{e^{\beta T}-1}{\beta }\\ 0 & 0 & 0 & e^{\beta T} \end{array}\right]x_{i,k-1}+\left[\begin{array}{c} 0\\ 0\\ \frac{g}{\beta }\left(\frac{e^{\beta T}-1}{\beta }-T\right)\\ \frac{g(e^{\beta T}-1)}{\beta } \end{array}\right]+w_{i,k},i=1,\ldots ,n \end{array} \end{array}$$
where $x_{i,k}={\left[p_{x},v_{x},p_{y},v_{y}\right]}^{T}$ denotes the state of the target *i* at the time *k*, containing the position and velocity in the direction of *x* and *y* axes. *T* indicates the sampling time interval, and Ω is the angular velocity in the CT model. In the DP model, β denotes the ratio of resistance coefficient to the mass of object, to simulate the damping effects, and *g* indicates the acceleration of gravity. $w_{i,k}$ is the target moving process noise with the 0-mean and $\sigma_{1}^{2}I_{4}$. The measurement equation is as follows: (21)$$\begin{array}{c} \begin{array}{c} z_{i,k}=\left[\begin{array}{cccc} 1 & 0 & 0 & 0\\ 0 & 0 & 1 & 0 \end{array}\right]x_{i,k}+v_{k},i=1,\ldots ,n \end{array} \end{array}$$
where $v_{k}$ is a 0-mean Gaussian noise with $\sigma_{2}^{2}I_{2}$ covariance.

For the distributed measurement system, the default setting is six measuring sensors. We use Lidar as the sensors that can only obtain the position information, and the measurement noise coefficient $\sigma_{2}^{2}$ is set to 5. In the tracking process, the number of tracked targets is set to 6, and the motion models taken by the targets were randomly configured. Since Lidar with a large detection range has low refresh frequency, the discrete time step is set to T = 1 s. The number of particles is set to 9, and the optimization solving accuracy ϵ is set to $10^{-6}$. Multiple simulation scenarios are generated by varying both the initial state of targets, the parameters of the motion models, and trajectory simulation time with each simulation spanning more than 20 discrete time steps. An example of the simulation configuration of the real trajectories scenario is shown in Figure 3. In this scenario, two targets are set to move with CV model, two targets are set to move with CT model, and two targets are set to move with DP model, and the discrete time steps are set to 20. For the motion model parameters, the angular velocity Ω is set to 0.1 rad/s, and the ratio of resistance coefficient β is −0.5 $s^{-1}$. The initial state of the six targets are configured as follows: ${\left[0,6,0,3\right]}^{T}$${\left[-10,3,5,6\right]}^{T}$ for the CV model, ${\left[50,0,0,10\right]}^{T}$, ${\left[100,0,0,10\right]}^{T}$ for the CT model, and ${\left[20,5,150,0\right]}^{T}$, ${\left[20,15,130,0\right]}^{T}$ for the DP model.

### 4.2. Determination of the Scaling Factor

The cost function includes two parts of errors: the coupled measurement errors and the error between particle prediction and fused measurements. The setting of the scale factor affects the effectiveness of the cost function. In order to determine the value of the scaling factor, we evaluate and compare the covariance of these two parts. According to the Equation (Equation 12), the expectation of the fused measurement error can be written as the product of the covariance of noise and the coefficient $\left(I+A_{m}^{T}A_{m}\right)$, since the measurement noises for sensors are the same. The first part has a fixed value compared to the covariance R and the second part depends only on the number of targets. The covariance of the fused measurements decreases as the number of targets increases and will converge to a limit value. Therefore, the ratio of fused measurement covariance to the direct measurement covariance decreases with the number of targets, and the result is shown in Figure 4. From the figure, we can see that the radio converges quickly as the number of targets increases, and the limit value of convergence is 0.4472. This result demonstrates that the fused measurements can achieve more than 50% noise reduction with the help of coupled measurement compared to direct measurement. Therefore, the scaling factor is positively proportional to the inverse of the noise variance; we have α:1−α = 0.4472:1, so α is set to 0.691 to accommodate most cases and to obtain the best performance. However, if the measurement noise of each sensor is different, the limit of scaling factor is difficult to calculate.

### 4.3. Analysis of Results 

For the evaluation of the tracking performance, we employ the of the target root mean square error (RMSE) between the tracking estimated position and the position ground truth as the evaluation metrics. The formulation of RMSE is shown as Equation (Equation 22), where *K* is the total time steps.(22)$$\begin{array}{c} RMSE=\frac{1}{K}\sum_{k=1}^{K}\sqrt{{\left(x_{i,k}-{\hat{x}}_{i,k}\right)}^{2}+{\left(y_{i,k}-{\hat{y}}_{i,k}\right)}^{2}} \end{array}$$

The performance of each algorithm is an average over 100 simulations. In a distributed system, there are various parameters that can effect the tracking performance, such as the spatial distribution of sensors, the signal transmission link of sensors, etc. For the spatial distribution of sensors, the geometric accuracy factor, for example, the Geometric Dilution of Precision (GDOP), is used to evaluate the accuracy for the target location. The GDOP can reach the best performance when the sensors are set up to form an equidistant polygon with the detected target in the center. We can also improve the positioning accuracy by increasing the number of sensors, but there will be a limit to improve the accuracy as the number increases. In this work, since we mainly discuss the effect of coupled measurement on target tracking accuracy, we assume that all sensors can measure each target, and the measurement error of sensors follows the same distribution. Different experiments have been designed to validate various performance characteristics and investigate the impact of various parameters in the proposed method.

(1)The comparison of tracking average RMSE.(2)The comparison of tracking RMSE over time.(3)The evaluation of measurement noise.(4)The evaluation of the number of targets.

We first compare the tracking performance of these algorithm. In the simulation, the system noise coefficient $\sigma_{1}^{2}$ is set to 0.1 and the measurement noise coefficient $\sigma_{2}^{2}$ is set to 5. The tracking results are averaged over 100 simulations and are shown in Table 1, and the results are also averaged over the targets with the same motion model. From Table 1, the experimental results demonstrate that the proposed distributed particle filtering method achieves superior tracking performance compared to both the conventional PF and the UKF across different motion models, with about 7%, 17%, and 16% RMSE reduction in CV, CT, and DP models, respectively, compared to the PF method’s performance. Since the targets on CV model are the most simple targets to be tracked, the performance gain is minimal. This comparative analysis confirms the effectiveness and robustness of the proposed approach. We also evaluate the program runtime for 100 simulations of these three methods, which are, respectively, 274.95 s 0.32 s, and 0.36 s. The proposed method is currently not suitable for real-time applications, but parallel computing skill, divide-and-conquer technologies, can be further implemented to reduce the computational time.

Then, we examine the tracking RMSE averaged over all targets versus time, and the results are shown in Figure 5. The tracking performance of the UKF, conventional PF, and the proposed distributed particle filter are denoted by a blue line with circular marker, a red line with a star marker, and a green line with a square marker, respectively. From the figure, we can see that the tracking error initially increases from zero before converging to a steady value over time. This occurs because the initial state of the target is assumed to be perfectly known, resulting in zero initial error. However, due to uncertainties in the system dynamics and measurement noise, the error accumulates before stabilizing. Upon reaching the steady RMSE, the proposed method consistently maintains a lower tracking error compared to both benchmark algorithms on different motion models, having more than 7% RMSE reduction compared to the PF method, demonstrating its enhanced estimation accuracy and stability.

Subsequently, the influence of measurement noise variations on tracking performance is systematically examined. We evaluate the algorithm using different measurement noise variances ranging from 0.5 to 50 (specifically [0.5,1,2,3,4,5,10,15,20,25,30,40,50]), while keeping other parameters fixed as in prior experiments. The experiments are also repeated 100 times, with target position errors averaged across all trials. The results of comparison for different tracking algorithms are shown in Figure 6 As shown in Figure 6, the proposed algorithm demonstrates superior tracking performance compared to both conventional PF and UKF approaches. Specifically, the the proposed algorithm achieves a lower average RMSE throughout the tracking duration while maintaining enhanced robustness to measurement noise. These results validate the effectiveness of incorporating coupled measurement into the conventional particle filtering framework.

Finally, we investigate the impact of the number of tracked targets *m* on algorithmic performance. The number of targets *m* is varied across the set [3, 5, 7, 9, 11, 13, 15, 17], while all other experimental parameters remain consistent with first configurations. For each value of *m*, 100 Monte Carlo simulations are conducted, and the tracking accuracy RMSEs across all targets are averaged for comparison. Figure 7 presents the resulting metrics for these evaluated algorithms, demonstrating their scalability with respect to the target count. As illustrated in the figure, after incorporating the distributed coupled measurement into the standard PF framework, the proposed method exhibits superior tracking accuracy, with a consistently lower average RMSE compared to both UKF and PF across the entire tracking duration. Moreover, the proposed method shows also a slightly lower rate of RMSE increase with the increasing number of targets: for example, on the DP model, the proposed method has a 10% RMSE reduction on 3 targets and it has a 18% RMSE reduction on 17 targets compared to the PF method. The experimental results demonstrate that the proposed method maintains robust multi-target tracking performance with lower degradation as the number of targets increases, confirming its scalability in complex and large tracked targets tracking scenarios.

To briefly summarize the different kinds experiments carried out, we first examined the proposed methods by comparing the tracking accuracy RMSE, and the results demonstrated that our proposed distributed particle filtering method outperforms the UKF and PF methods. Our method achieved a trackin performance gain of more than 7% for various moving models. Then, we examined the robust against the measurement noise, the results show that the proposed method maintains the best performance, showing enhanced robustness to measurement noise. Finally, we examined the performance on various numbers of tracked targets. With an increasing number of targets, the proposed methods outperform the other methods, and their tracking accuracy RMSE increases slightly slower with an increasing number of targets. The result justifies the capability of the proposed method for complex and numerous multi-target tracking problems. With this assumption, we show that the sensors can detect all the targets and the noises are identical, which is not realizable in a realistic scenario. In a realistic scenario, the imperfect communication links will lead to inconsistent measurement noise and a different time delay. The different time delay can be compensated partially, since the time step for Lidar with a large detection range is long and can also increase measurement noise. The target loss and false positives have a direct impact on the data association, incorrect data association degrades the tracking of the target state, which in turn increases the tracking error. Further research is still needed on how to perform data association and to evaluate the impact of inconsistent measurement noise.

## 5. Conclusions

In this paper, we propose a novel distributed particle filter with coupled measurements for multi-target tracking systems. Based on conventional particle filtering, we introduce distributed coupled measurements and derive fused measurements through the weighted least squares estimation method, thus maximizing measurement utility while reducing measurement system noise. We build a cost function to comprehensively integrates direct measurements, coupled measurements, and fused measurements to optimize particle weight distribution via constrained optimization. The optimized particle weight estimation enables more robust state estimation, demonstrating improved resistance to both measurement noise and scalability with an increasing numbers of targets. Our analysis examines different critical performance factors: the model variations, measurement noise, and the number of targets. Simulations on randomly generated trajectories demonstrate that the distributed particle filter outperforms other methods such as the conventional PF and UKF, reducing average RMSE by more than 7% compared to the PF method, the second best method. There is still a lot of work that needs to be continued in order to apply the proposed algorithm in a practical way. Firstly, the algorithm is not real-time, so a simpler and more efficient algorithm is still wanted. Future work will focus on computational optimization for mobile platform deployment, facilitating large-scale surveillance network implementation. Secondly, the noise of the system is inconsistent in real scenarios and has difficulty with analytic analysis, so the varied tracking effects need to be studied to improve the system robustness. Finally, for multi-target tracking, the data association strategy has an important impact on tracking performance, so further research on a robust data association strategy is still needed.

## Figures and Tables

**Figure 1 sensors-25-03495-f001:**
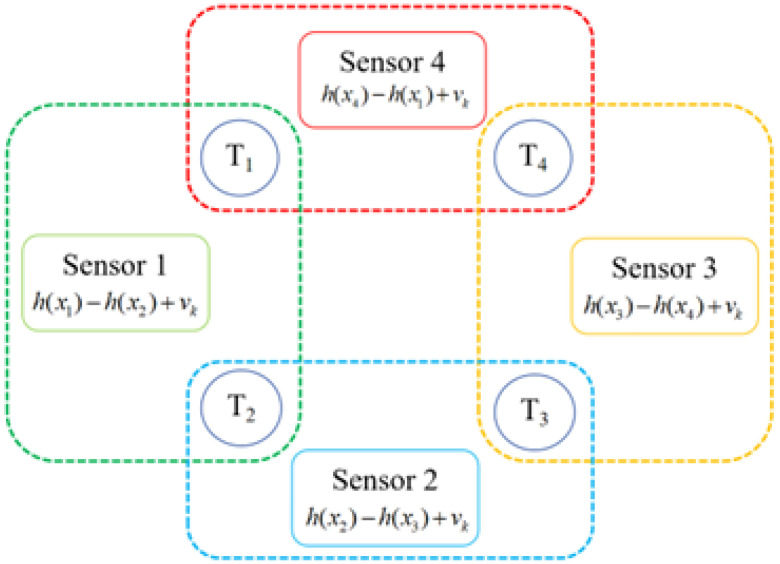
Schematic diagram of obtaining relative state measurement information.

**Figure 2 sensors-25-03495-f002:**
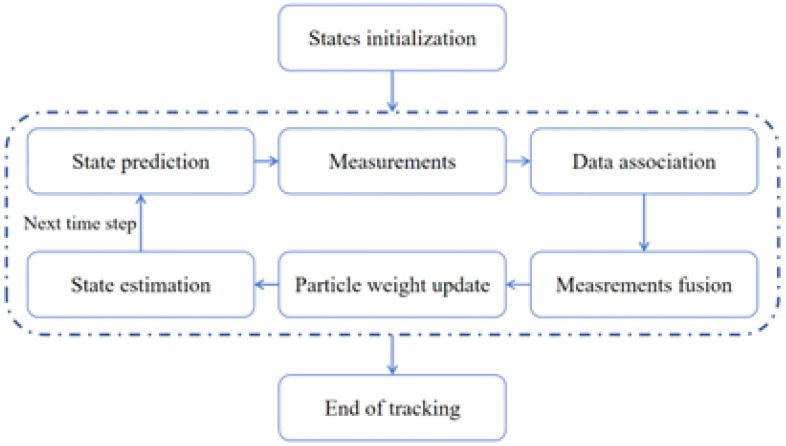
Flowchart of distributed particle filter multi-target tracking algorithm.

**Figure 3 sensors-25-03495-f003:**
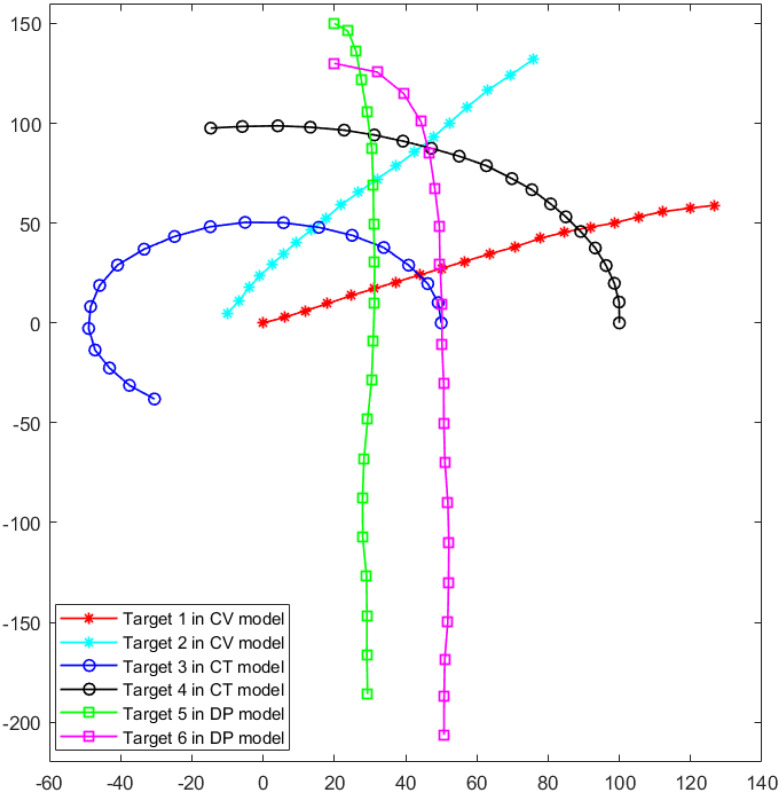
An example of the real trajectories scenario with 6 targets on different moving models.

**Figure 4 sensors-25-03495-f004:**
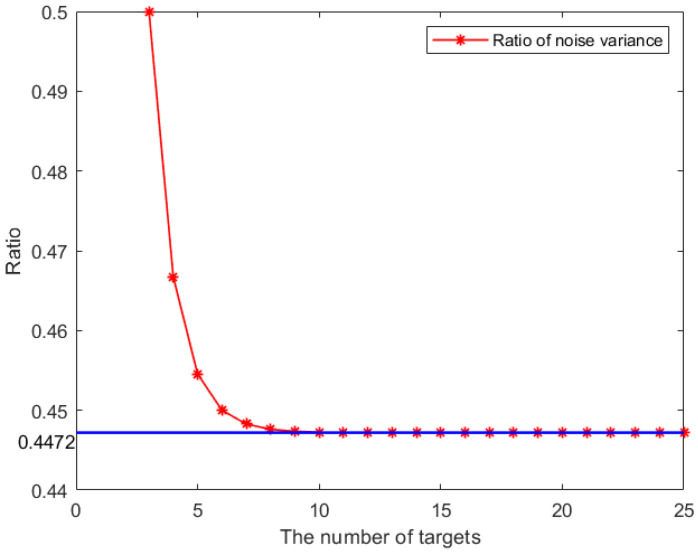
The ratio of of the noise variance with increasing number of targets.

**Figure 5 sensors-25-03495-f005:**
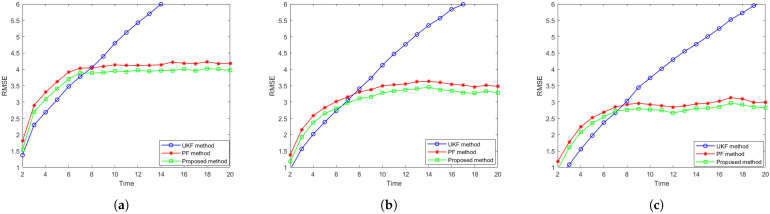
Comparison of tracking accuracy RMSE over time for (**a**) the CV model, (**b**) the CT model and (**c**) the DP model.

**Figure 6 sensors-25-03495-f006:**
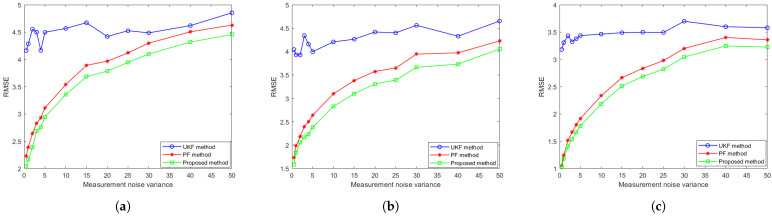
The evolution of tracking accuracy RMSE over measurement noise for (**a**) the CV model, (**b**) the CT model and (**c**) the DP model.

**Figure 7 sensors-25-03495-f007:**
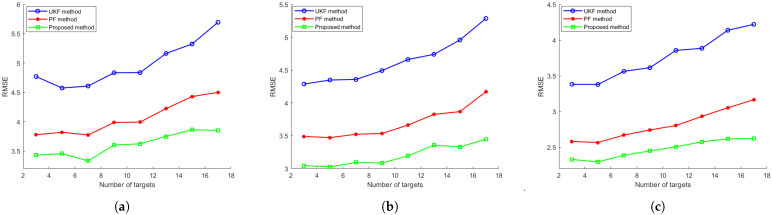
The evolution of tracking accuracy RMSE over the number of targets for (**a**) CV model, (**b**) the CT model and (**c**) the DP model.

**Table 1 sensors-25-03495-t001:** Comparison of tracking RMSEs.

Model	PF (m)	UKF (m)	Proposed Method (m)
CV model	4.75	6.81	4.43
CT model	3.21	5.72	2.68
DP model	2.71	3.91	2.29

## Data Availability

The data presented in this study are available on request from the corresponding author. The data are not publicly available due to privacy.

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
