# Peer review of "Research on a Particle Filtering Multi-Target Tracking Algorithm for Distributed Systems"

_sensors, 2025, doi:10.3390/s25113495_

Round 1

Reviewer 1 Report

Comments and Suggestions for Authors
  1. When some abbreviations first appear, they need to be given their full description, such as UAV, MIMO, etc.
  2. For distributed systems, the spatial distribution of sensors has a significant impact on performances, and there is no analysis or discussion on this in the manuscript.
  3. In Figure 1, the rightmost sensor 4 should be sensor 3.
  4. The vectors and scalars in the text should be written differently.
  5. For multi-target tracking scenarios, data association is an important processing step, and its performance is often more affected than that of filters. This manuscript also has almost no discussion on specific data association methods.
  6. Why is the text in the diagram tilted in the middle of Figure 2?
  7. In Section 4, there is almost no explanation about simulation parameters, such as target position, motion parameters, sensors type, error, etc.
  8. How is the value of RMSE calculated? Is it a statistical analysis of the tracking results for all targets? How can the performance differences of different motion model targets be reflected?

Reviewer 2 Report

Comments and Suggestions for Authors

The work is devoted to one of the methods of Kalman filtering, which allows improving the performance of the Kalman filter in a situation with many goals.

The introduction to the work gives an adequate idea of ​​the state of affairs in the field of research and the problems that the authors solve.
The goals of the study, the purpose of the work and the novelty that the authors contributed are correctly stated.

The work consistently and clearly sets out the methodology for solving the problem and clearly states the results obtained.

I have no questions about the figures.

I think the work can be published in its current form.

Reviewer 3 Report

Comments and Suggestions for Authors

Key remarks and recommendations:

  1. While the introduction provides an overview of existing works, a more comprehensive comparison of the proposed algorithm with state-of-the-art distributed tracking algorithms could be beneficial. This would better contextualize the contribution of this work within the current research landscape.
  2. Additionally, a point of concern arises in the text where it states, "where Is denotes the unit matrix and s denotes the dimension of the measurement of one sensor. From Eq. 12, we can obtain:," immediately preceding the mathematical expression (8). This reference to "Eq. 12" appears to be inaccurate, as the subsequent mathematical derivation seems to follow logically from the definitions and relationships established in the preceding equations, particularly Equations (1) and (2), rather than Equation (12). This discrepancy in the equation referencing warrants clarification to ensure the logical flow and traceability of the presented derivations.
  3. A more in-depth justification is warranted for the assumption of particle weight independence, as presented in Equation (14). The assertion that the weight of a particle pair for a coupled measurement is simply the product of their individual weights may not fully capture potential correlations between target states. Further explanation is needed on how this assumption influences the algorithm's ability to adequately handle scenarios with complex inter-target dynamics.
  4. While the paper formulates the particle weight optimization problem in Equations (16) and (17), it lacks sufficient detail regarding the implementation of the solution method. It would be beneficial for readers to understand the specific optimization algorithm employed, its parameter settings, and its computational demands. Insight into the computational complexity of this stage is crucial for assessing the practical applicability of the proposed approach.
  5. A discussion on the overall computational complexity of the proposed algorithm in comparison to the baseline methods, such as conventional particle filtering and unscented Kalman filtering, is necessary. The additional steps of measurement fusion and weight optimization could significantly increase the computational burden, which may present limitations for real-time applications, particularly with a large number of targets.
  6. The limitations of the conducted simulations should be more clearly delineated. For instance, the paper does not consider realistic scenarios such as imperfect communication links between sensors, asynchronous measurements, target loss, or the presence of false positives. An analysis of how these factors might impact the performance of the proposed algorithm would enhance the practical relevance of the study.
  7. In the review process, careful examination of Figures 3 and 4, illustrating the trajectory simulation scenario and the covariance dependency justifying the choice of parameter α, is essential. A thorough understanding of the information presented in these figures is crucial for evaluating the adequacy of the simulation setup and the validity of the adopted decisions. (As I lack access to visual content, I cannot assess them directly).
  8. A discussion of the potential challenges and specific considerations for the practical implementation of the proposed algorithm in real-world UAV systems, such as computational resource requirements, data transmission latencies, and robustness to sensor failures, would significantly enhance the value of the paper for engineers and developers.

Round 2

Reviewer 1 Report

Comments and Suggestions for Authors

My comments have been addressed.